# Potential Tear Biomarkers for the Diagnosis of Parkinson’s Disease—A Pilot Study

**DOI:** 10.3390/proteomes10010004

**Published:** 2022-01-13

**Authors:** Arantxa Acera, Juan Carlos Gómez-Esteban, Ane Murueta-Goyena, Marta Galdos, Mikel Azkargorta, Felix Elortza, Noelia Ruzafa, Oliver Ibarrondo, Xandra Pereiro, Elena Vecino

**Affiliations:** 1Experimental Ophthalmo-Biology Group, Department of Cell Biology and Histology, University of the Basque Country UPV/EHU, 48940 Leioa, Spain; aacera71@gmail.com (A.A.); noelia.ruzafa@ehu.eus (N.R.); xandra.pereiro@gmail.com (X.P.); 2Department of Neuroscience, Biodonostia Health Research Institute, 20014 Donostia-San Sebastian, Spain; 3Department of Neurology, Cruces University Hospital, 48903 Barakaldo, Spain; juancarlos.gomezesteban@gmail.com; 4Neurodegenerative Diseases Group, BioCruces Health Research Institute, 48903 Barakaldo, Spain; ane.muruetagoyena@gmail.com; 5Department of Neurosciences, Faculty of Medicine and Nursery, University of the Basque Country UPV/EHU, 48940 Leioa, Spain; 6Begiker-Ophthalmology Research Group, BioCruces Health Research Institute, 48903 Barakaldo, Spain; marta.gal2@gmail.com; 7Department of Proteomics, Center for Cooperative Research in Biosciences, 48160 Derio, Spain; mazkargorta@cicbiogune.es (M.A.); felortza@cicbiogune.es (F.E.); 8RS-Statistics, 20500 Arrasate, Spain; oibarrondo@rs-statistics.com

**Keywords:** biomarkers, Parkinson’s disease, tear film, lysosome

## Abstract

Parkinson’s disease (PD) is the second most common neurodegenerative disease after Alzheimer’s disease. In this study, the tear proteome profile of patients with idiopathic PD (iPD, *n* = 24), carriers of the E46K-SNCA mutation (*n* = 3) and healthy control (CT, *n* = 27) subjects was analyzed to identify candidate biomarkers for the diagnosis of PD. An observational, prospective and case-control pilot study was carried out, analyzing the participants tear samples by nano-liquid chromatography–mass spectrometry (nLC–MS/MS) and assessing their neurological impairment. The proteomic data obtained are available at ProteomeXchange with identifier 10.6019/PXD028811. These analyses led to the identification of 560 tear proteins, some of which were deregulated in PD patients and that have been implicated in immune responses, inflammation, apoptosis, collagen degradation, protein synthesis, defense, lipid transport and altered lysosomal function. Of these proteins, six were related to neurodegenerative processes and showed a good capacity to classify patients and controls. These findings revealed that certain proteins were upregulated in the tears of PD patients, mainly proteins involved in lysosomal function. Thus, in this study, tear proteins were identified that are implicated in neurodegeneration and that may be related to an aggressive disease phenotype in PD patients.

## 1. Introduction

Parkinson’s disease (PD) is the second most common neurodegenerative disease after Alzheimer’s disease (AD), and it is characterized by slowness of movement (bradykinesia), increased muscle tone (rigidity), shaking (tremor) and impaired postural control. In pathologically confirmed PD, there is a variable amount and deposition of misfolded alpha-synuclein (α-syn) in the form of Lewy bodies (LBs) and Lewy neurites in the central nervous system (CNS). This variability in the accumulation of these structures is likely to reflect the clinical heterogeneity in this disease [1,2,3]. Indeed, several clinical disorders lie in the spectrum of Lewy body diseases (LBDs), including patients with idiopathic (iPD) and genetic PD and patients suffering dementia with LBs. One of the main challenges in dealing with neurodegenerative diseases is to find clinical markers that allow for the early classification of patients and help to monitor disease progression.

Factors such as oxidative stress, proteolysis, altered fatty acid or phospholipid concentrations have been associated with structural changes to α-syn, and post-translational modifications can also alter the size, structure or protein load of α-syn [4]. Deregulating mutations and multiplications in the α-syn gene (SNCA) cause autosomal-dominant familial forms of PD [5]. Perhaps the most pathogenic mutation known to induce PD is the E46K mutation in α-syn (E46K-SNCA), which underlies a group of aggressive, genetic LBDs with clinical and pathological features reminiscent of dementia with LBs [6,7]. Although mutations in α-syn are rare, they provide a unique opportunity to identify patients with PD and to better understand the progression of these pathologies.

It is known that PD patients develop hallucinations, changes in eye and eyelid movement, and altered tear composition and in reduced amounts [8,9]. In addition to eye-surface damage, these patients can experience changes in accommodation, reduced visual acuity, scotoma formation (areas where their field of vision is partially diminished or completely degenerated) and a thinning of the retinal layers, particularly due to a reduction in the number of nerve fibers [10,11]. Deficiencies in neurotransmission and monoamine metabolism also influence the pathological changes to the visual system of PD patients [12]. These changes are quite typical, since monoamines are involved in the transmission of visual information in the retina [13]. Moreover, PD patients may experience alterations to the innervation of the anterior part of the eye that can ultimately alter the composition of tears.

Many neurotransmitters and trophic substances are secreted in tears, originating from the aqueous humor or from nerves running through the cornea [14]. However, the cellular and molecular mechanisms that underlie these associations have yet to be fully elucidated. As PD affects several non-motor systems and peripheral nerves, lacrimal secretion could be altered in these patients, and the tear protein composition might exhibit a characteristic profile in these patients that could serve as a diagnostic biomarker. In this sense, analyzing the tear proteins of patients with E46K-SNCA and iPD could help to identify useful candidate biomarkers for the diagnosis and progression of PD. Therefore, the aim of the present study was to analyze the tear proteome profile in iPD patients and carriers of the E46K-SNCA mutation in order to find a biomarker that could help in the future diagnosis of PD, using non-invasive samples.

## 2. Materials and Methods

### 2.1. Study Cohort

We designed an observational, prospective, case-control pilot study in which 24 patients with iPD, 3 carriers of the E46K-SNCA mutation and 27 healthy subjects (CTs) were enrolled. This research was conducted by medically qualified personnel after receiving approval from the OSI Ezkerralde-Enkaterri-Cruces Ethics Committee CEIC E18/47. The study was carried out in strict accordance with the tenets of the Helsinki Declaration on Biomedical Research Involving Human Subjects. Before sample collection, signed informed consent was obtained from all the subjects after having explained the nature and possible consequences of the study. The patients were recruited at the Neurology and Ophthalmology Service Units of the Cruces University Hospital (Barakaldo, Bizkaia, Spain) during outpatient consultations, based on the agreed inclusion and exclusion criteria. Patients with iPD fulfilled the Parkinson’s UK Brain Bank criteria for the diagnosis of PD and did not carry any known mutations. The inclusion criteria for CTs were to be over 40 years of age, of either gender and without a clinical history of any neurological disorder. Individuals with (or with a history of) any systemic or ocular disorder/condition were included in the study. The exclusion criteria consisted of eye surgery in the previous three months, chronic eye medication (e.g., for glaucoma), a history of allergies, using any topical medication (other than artificial tears) or oral medication containing corticosteroids, atopy and patients with Sjögren’s syndrome. Contact-lens users were also excluded to avoid any possible interference with the interpretation of the results. Preliminary ocular tests and sample collection were performed on the same day, while patients with PD were also seen on a separate day by members of the research team who are experts in movement disorders to assess the Parkinson’s Scales.

### 2.2. Ophthalmological Examination

The clinical ophthalmological examination included the Schirmer’s test I (SCH), which determines whether the eye produces enough tears to keep it moist. The test allows the water in tears to travel along the length of a paper test strip, and normal values are ≥10 mm wetting of the paper after 5 min. Tear-film instability was assessed by using tear breakup time (TBUT) with fluorescein. TBUT is the time taken for the first dry spot to appear on the cornea after a complete blinking. A value less than 10 s usually reflects some tear-film instability. Any systemic disease or any other condition/medication use that could interfere with the interpretation of the results was considered a criterion for exclusion.

### 2.3. Neurological Examination

A neurologist expert in the field of movement disorders recorded the age of the PD patients at tear collection, their disease duration and their score on the Unified Parkinson’s Disease Rating Scale (UPDRS) and Hoehn Yahr scale. These tests serve as stratification criteria to track the progression of the disease. The confidentiality of patient data was ensured through the use of the Basque Biobank (www.biobancovasco.org (accessed on 2 November 2019).

### 2.4. Tear-Sample Collection

All tear samples were collected by using calibrated 10 µL glass microcapillary tubes (BLAUBRAND intraMark, Wertheim, Germany). Tear samples were obtained from the inferior temporal tear meniscus, minimizing irritation of the ocular surface or lid margin, and without installation of anesthesia. The tear samples were collected from both eyes of each participant and immediately placed in precooled Eppendorf tubes. After tear collection, the samples were stored at −80 °C until their analyses.

### 2.5. Proteomics Analyses

The proteomic analyses were carried out at the CIC bioGUNE Proteomics Service (Derio, Bizkaia, Spain), using the Filter Aided Sample Preparation (FASP) protocol [15] for sample processing and digestion. Trypsin (Sigma Aldrich, St. Louis, MO, USA) was added at a trypsin:protein ratio of 1:50, and the mixture was incubated overnight at 37 °C, dried in a RVC2 25 Speedvac concentrator (Christ, Vienna, Austria). The peptides obtained were desalted and resuspended in 0.1% Formic Acid (FA) (Sigma Aldrich, St. Louis, MO, USA) using C18 stage tips (Millipore, St. Louis, MO, USA).

Samples were analyzed in a novel hybrid trapped ion mobility spectrometry quadrupole time of flight mass spectrometer (timsTOF Pro with PASEF, Bruker Daltonics, Bremen, Germany), coupled online to a nanoElute liquid chromatograph (Bruker, Coventry, UK). This mass spectrometer takes advantage of a novel scan mode termed parallel accumulation serial fragmentation (PASEF), which multiplies the sequencing speed without any loss in sensitivity, and it has been proven to provide outstanding analytical speed and sensitivity for proteomics analyses [16]. Samples (200 ng) were directly loaded onto a 15 cm Bruker nanoelute FIFTEEN C18 analytical column (Bruker, Coventry, UK) and resolved at 400 nL/min. The column was heated to 50 °C in an oven.

Protein identification and quantification were carried out by using the PEAKS software (Bioinformatics solutions Inc., Waterloo, CA, USA) Searches were carried out against a database of canonical human Uniprot/Swissprot entries (no isoforms considerer), with precursor and fragment tolerances of 20 ppm and 0.05 Da. An area-based label-free protein quantification was performed by using the PEAKS Q module available in the PEAKS software. Only proteins identified with at least two peptides at a False Discovery Rate (FDR) < 1% and present in at least 70% of the samples from one of the experimental groups under analysis were considered for further analysis. The data were loaded onto the Perseus platform [17] and further processed (log2 transformation, imputation) before the application of a Student’s *t*-test for differential protein-expression analysis. The proteomics data are summarized in Appendix A, and the mass spectrometry proteomics data were deposited to the ProteomeXchange Consortium via the PRIDE [18] partner repository, with the dataset identifiers PXD028811 and 10.6019/PXD028811.

### 2.6. Statistical Analyses

Statistical analyses were carried out by using the IBM SPSS Statistics package for Windows, 23.0v (IBM-SPSS, Armonk, New York, NY, USA). The normality of continuous data was assessed with a Shapiro–Wilk test. The *p*-values calculated determine the probability that the association between proteins in the dataset and a given canonical pathway, functional network or upstream regulator is explained by chance alone, based on a Fisher’s exact test, with a *p*-value < 0.05 being considered significant.

Univariate logistic regressions were used to identify the statistically significant characteristics associated with PD-patient discrimination. After variable selection, multivariate logistic regressions were used to construct a PD discrimination model and to identify the clinical and proteomic variables relevant for PD identification. Logistic models allow the relationship between a discrete dependent variable and a set of continuous or discrete independent variables to be estimated. The models allow the classification of patients as with or without PD, calculating the probability of belonging to each group. The *p*-value of each covariate describes the statistical significance of the association between the response and each term included in the model.

## 3. Results

### 3.1. Patients and Clinical Parameters

Between March 2019 and October 2019, 24 patients with iPD, three symptomatic carriers of the E46K-SNCA mutation and 27 CT subjects were included in this pilot study, whose general characteristics are presented in Table 1. The PD patients’ mean disease duration was 9.36 years, and they had mild bilateral motor disability (UPDRS score) and Hoehn and Yahr scores that suggested mild-to-moderate motor impairment. A characteristic of iPD patients was that 40% of them had mild-to-moderate blepharitis.

### 3.2. nLC MS/MS Data

A total of 560 tear proteins were identified in the samples analyzed here, in accordance with previous proteomic studies on tears, and several group-specific tear proteome alterations were evident in the PD patients relative to the CTs. Proteins with different abundances were detected in both groups, although most of the deregulated proteins in the tear proteome of PD patients were overexpressed. An analysis of the STRING PPI network was performed to evaluate the interactions between proteins that differed significantly in the PD patients and CTs (Figure 1). Proteins that were overexpressed in PD patients were implicated in significantly more interactions than expected, indicating that the increases in those proteins have some biological connection (Figure 1A). However, weaker relationships were observed between the proteins that were downregulated, representing a clearly differentiated cluster (Figure 1B). Most of the differentially expressed proteins could be localized to either the extracellular and intracellular compartments.

To better understand the potential role of the differentially expressed proteins in a biological context, functional annotation analyses were performed with different gene ontology (GO) terms. As a result, deregulated proteins were seen to be involved in different biological processes (Figure 2): the immune response (lipocalin 2—LCN 2, serpin B3—SPB3, follicular dendritic cell secreted peptide—FDSCP, lactadherin—MFGM and neutrophil gelatinase-associated lipocalin—NGAL); apoptosis (olfactomedin-4—OLFM4, caspase-14—CASPE and gamma-glutamylcyclotransferase—GGCT); inflammation (Calprotectin-S100A8, S100A9); collagen degradation (myeloblastin—PRTN3 and metalloproteinase 9—MMP9); protein synthesis (heterogeneous nuclear ribonucleoprotein A3—ROA3); keratinocyte differentiation (calmodulin-like protein 5—CALML5); lipid transport (apolipoprotein D—APOD, apolipoprotein A2—APOA2 and low-density-lipoprotein-related protein 2—LRP2); and defense (myeloperoxidase—MPO, defensin alpha 3—DEFA 3, glutathione peroxidase 3—GPX3, chitinase-3-like protein 2—CH3L2 and lysozyme C—LYSC).

If the PD patients were considered all together, and separately as iPD patients and patients with the E46K-SNCA mutation, upregulated tear proteins were evident in all three of these groups, with their expression varying in each (Figure 3). It should be noted that only three patients carrying the E46K-SNCA mutation were studied, and the dispersion of the data from then was greater. The same proteins were more strongly expressed in the tears of patients with the E46K-SNCA mutation than in the tears of patients with iPD. All of these comparisons were made on the values relative to those of the CT tear samples.

Table 2 shows the proteins that were upregulated (fold > 1.5) and downregulated (fold < 0.5) in the iPD group with blepharitis when compared with iPD patients without blepharitis. Patients with the E46K-SNCA mutation did not develop blepharitis.

Among the tear proteins deregulated in PD patients, a group of six proteins stood out, five of which that were overexpressed in the PD group relative to the CTs (prelamin A/C—LMNA, cathepsin D—CATD, acid ceramidase—ASAH1, transitional endoplasmic reticulum ATPase—TERA and cytoplasmic dynein 1—DYHC1), whereas one of them was downregulated (tripeptidyl-peptidase 1—TPP1: Table 3). These proteins were associated with neurodegenerative processes, mainly alterations in lysosomal autophagy, apoptosis, retrograde axonal transport and demyelinating processes. It was interesting to note the relationship of these proteins with alterations in lysosomal function. Of the six proteins, three presented a good capability to correctly classify PD patients and CTs, as reflected by the AUC values of the ROC curves, and they were related to alterations in lysosomal function: CATD, ASAH1 and DYHC1.

The univariate logistic regression analysis identified four variables with statistical significance at α = 0.05: the age of the patients and the fold change in CATD, ASAH1 and DYHC1 proteins (Table 4). 

In addition, the LMNA and DYHC1 proteins were seen to have statistical significances at α = 0.1. A multivariate logistic regression model was established by considering age and proteins with any significance level. While the general model only showed significance in the case of the CATD protein (Table 5), the model was likely to have been affected by the small number of samples available, and increasing the number of samples studied might well identify other significant variables.

## 4. Discussion

The present study was carried out to analyze proteins by nano-liquid chromatography–mass spectrometry (nLC–MS/MS) and select those related to neurodegenerative diseases that had a higher discriminatory power between the PD and TC groups studied. Several searches for candidate biomarkers in the tears of PD patients have been performed in recent years; however, the conclusion was that further validation studies were needed to find the best candidate diagnostic or prognostic biomarker for this pathology. The diagnosis of PD is currently based on the presence of motor and non-motor symptoms, including sleep disturbances or olfactory deficits. Motor symptoms appear mainly when the loss of dopaminergic neurons reaches around 50–60%. Therefore, achieving an early diagnosis will open the way to modify the course of the disease and hopefully delay the severe disability caused by this pathology. This issue has become a key aspect in PD research, although the lack of valid biomarkers represents a major obstacle that prevents the identification of patients in preclinical or prodromal stages, as well as the monitoring of the evolution of the disease and the effects of treatments. For this reason, the present study set out to search for a reliable biomarker of PD in a noninvasive fluid, such as tears. The eye has many neural and vascular elements that are also found in the brain, making it ideal for the discovery of new biomarkers that could be used to diagnose PD or other neurodegenerative diseases, as well as in the discovery of new therapeutic targets.

Here, tears from PD and CT patients were obtained through glass capillaries without prior anesthesia and without touching the ocular surface. Previous pilot studies on patient tears have been performed with a group of patient tears [19], and our analysis is the first individual tear analysis that will allow for the identification of specific individual markers. By using capillaries to obtain the tear samples, we avoid touching the conjunctival epithelium, thus discriminating proteins found in the tear from conjunctival proteins, as occurs with the use of Schirmer strips.

In our study, we found significantly deregulated proteins in PD patients. These proteins are mainly involved in inflammatory and neurodegenerative processes, apoptosis and immune responses. Inflammation is ubiquitous in neurodegenerative diseases, including PD, and recent studies have identified that α-syn-derived T-cell epitopes are preferentially recognized in PD patients, as well as T cells in PD target regions, possibly suggesting an autoimmune component of PD [20]. Neurons were thought to be protected from autoimmune attack, but dopaminergic neurons are vulnerable because they have proteins on their cell surface that help the immune system recognize foreign substances [21]. T cells can potentially mistake disease damaged neurons for foreign substances, and, as a result, Sulzer et al. proposed a model in which α-syn-specific T cells cause neuronal death in neurodegenerative diseases associated with specific misfolding of this protein. In our study, groups of deregulated proteins related to immune response, such as LCN2, SPB3 and FDSCP, among others, were observed in tear samples. However, certain immunoglobulins involved in immune response, such as IgA2, did not present statistically significant differences between the PD group and the CT group. Further studies are needed to elucidate whether this relationship between immune response and proteins found in tears is directly related to PD.

Neuroinflammation can be defined as a non-specific inflammatory event in the brain, and the entire CNS has a neuroinflammatory component, which is evident in diseases such as Multiple Sclerosis (MS), brain aneurysms and cerebrovascular accidents, epilepsy, AD and PD, and in which the activity of extracellular matrix (ECM) degrading enzymes or metalloproteases (MMPs) has been implicated. MMPs participate in many physiological and pathological processes in the brain and blood. The blood–brain barrier (BBB) is formed by the endothelial capillaries that separate the blood supply to the brain, and its function resides in three structures that are critical to its integrity: the endothelial cells of the brains capillaries, the tight junctions (TJs) between these cells, and the basement membrane. The endothelium is a barrier to small hydrophilic compounds as the TJs seal the gaps between the adjacent endothelial cells, impeding the uncontrolled passage of solutes between these cells and converting the brain endothelium into a relatively impermeable barrier [22,23]. However, the basement membrane connects these endothelial cells to pericytes and astrocytes to form the neurovascular unit that facilitates communication between the cells. This membrane is fundamental for the BBB to function adequately and, as such, for brain homeostasis and its general health. It has been proposed that MMPs specifically disrupt TJs, and they digest the basement membrane of the endothelium, thereby contributing to certain brain diseases [24]. While it is technically difficult to demonstrate MMP activity in vivo, stronger MMP activity was associated with greater BBB permeability following a cerebrovascular accident and during reperfusion in vivo [25]. An increase in MMP-2 and MMP-9 mRNA and activity has been described after reperfusion in hypertensive rats with medial cerebral artery occlusion (MCAO) [26]. Moreover, BBB leakage was also detected in the piriform cortex of these rats, in conjunction with the disruption of TJs, indicating that these MMPs alter BBB integrity by degrading TJ proteins [26]. As such, and despite the technical challenges, there is some evidence that MMPs digest TJ and ECM proteins in vivo. The expression and activity of MMP-1, -2 and -9 have been determined in postmortem brain tissue from PD patients, and while no changes were observed in MMP-1 and MMP-9 relative to aged-matched controls, a 50% reduction in MMP-2 activity was evident in the SN of the PD patients [27]. It was proposed that the weaker MMP 2 activity could help the dopaminergic neurons and their neurites survive in the SN [27]. Moreover, MMP-3 has also been studied in relation to PD, which could potentially be implicated in the cleavage of α-syn. MMP-3 cleaves purified α-syn in vitro and the aggregation of α-syn augments in the presence of MMP-3 aggregates that would be particularly damaging to dopaminergic cells in PD. Significantly, MMP-1, -2 and -9 and MT1-MMP also cleave purified α-syn, albeit less efficiently than MMP-3 [28]. MMP-3 appears to be involved in dopaminergic neurodegeneration, neuroinflammation and BBB integrity in PD; however, further studies will be required to clarify the role of all of these MMPs in PD and whether their specific inhibition could represent a valid therapeutic strategy. Here, MMP-9 levels were higher in the tear of PD patients, both iPD patients (2.8-fold) and those that carry the E46K-SNCA mutation (3-fold). The latter PD patients did not develop blepharitis, thus suggesting that the increase in MMP-9 concentrations is not as strongly associated with blepharitis as with the pathology of PD. However, analyzing the iPD groups separately, we see that there is stronger MMP-9 overexpression in iPD patients with blepharitis, 3.08 times that in iPD patients without blepharitis. We previously studied the contribution of this MMP to the tear in patients with pathologies of the ocular surface [29,30]. However, further studies will be necessary to determine if the increase in MMP-9 is directly related to PD or ocular surface pathologies that could alter the tear analysis.

Evidence is accumulating on the important role of energy metabolism in neurodegenerative diseases, including PD. Abnormal lipid fractions and lipid peroxidation have been assessed in PD patients [31], and, as a result, lipid fractions were proposed as possible protective biomarkers of PD. Recent epidemiological studies on a prospective cohort or with case-control designs attempted to assess the associations between lipid fractions and the risk of PD [32,33,34,35]. Some of these studies had small sample sizes, while others only examined specific lipid fractions, mainly cholesterol, or controlled for a limited number of potential confounders. Although a consistent association has been seen between higher cholesterol levels and a lower risk of PD, the underlying cause is unknown, and whether similar results are seen for other lipid fractions should be assessed.

In the tear samples tested here, members of the apolipoprotein (APO) family appear to be deregulated, a family of proteins implicated in many neurodegenerative disorders. The APOE protein participates in several brain disorders, including AD, MS, traumatic brain lesion and Creutzfeldt–Jakob disease [36,37,38], raising the possibility that ApoE may also play a significant role in the development of PD. Furthermore, higher cholesterol levels were detected in the brain of a transgenic mouse model of PD, and since APOs are involved in maintaining cholesterol homeostasis, a possible association between APOs and PD could exist [39]. However, few studies have focused on the role of APOs in PD. In the CNS, APOs are produced by glial cells and they are taken up by neurons to be used in axon growth and for synaptic activity. Furthermore, ApoE is thought to have a neuroprotective role, preventing the apoptosis of neurons and retinal ganglion cells (RGCs) [40]. The expression of APOs in the human brain is influenced by the type of APO and the individual’s age. Of the different APOs, ApoE, ApoD and ApoJ are the most strongly expressed in the brain. However, their levels of expression could change at different stages of life and, for example, there is 50% more ApoE in the neonatal brain than in that of an adult, while there is approximately 10 times more ApoD and ApoJ in the adult brain than in that of the embryo. ApoD is expressed principally in the brain, peripheral nerves, placenta, lung, ovary and spleen [41]. In our tear samples, ApoD appears to be overexpressed in patients with PD. Glial cells close to the SN of PD patients have more ApoD, which is thought to be related to oxidative stress, as ApoD apparently protects cells from oxidative stress, and its enhanced expression inhibits lipid peroxidation [42]. Indeed, ApoD’s expression in the brainstem appears to protect against neurodegeneration, and in damaged brains, neurons do not express ApoD [41]. Thus, ApoD in the tear warrants further study in PD patients and those with other neurodegenerative diseases to assess whether it has a true neuroprotective effect in such patients.

Among other proteins found to be deregulated in the present study, a small group was selected due to their involvement in neurodegenerative processes, specifically in lysosomal autophagy or neuronal transport. The ubiquitin proteasome (UPP) pathway has been proposed to play a key role in the degradation of α-syn [43]; however, there is growing evidence that the lysosome may also mediate the degradation of α-syn [44,45]. Regardless of the exact autophagy pathway by which α-syn enters the lysosome, it is assumed that it is rapidly degraded by “synucleinase” under normal conditions. In dopaminergic neurons, there is a dynamic equilibrium between the various conformational forms and oligomeric states of the α-syn protein, which is modulated by factors that can accelerate or inhibit the aggregation and formation of fibrils [46]. The identification and characterization of the toxic α-syn species is still incomplete, and many studies have focused on different states of protein aggregation. These studies attempted to determine if the toxic species correspond to the insoluble fibrillary proteins found mainly in the LBs or, conversely, to the pre-fibrillary protein oligomers or protofibrils. There is increasing evidence in vivo and in vitro that oligomeric species are the most pathologically relevant isoforms [47,48], and it has even been proposed that LBs could be protective, representing a form of aggresome. In physiological circumstances, α-syn is active at synapses and it is involved in processes such as the formation, trafficking and coupling of synaptic vesicles (SVs). It is also associated with the recycling of SVs and the storage of dopamine. The phosphorylation and dephosphorylation of α-syn drives the activation and deactivation of this protein, which is also controlled by lysosomal autophagy and proteasome degradation driven by ubiquitinylation [49]. In pathological conditions provoked by different stimuli, α-syn undergoes incorrect folding, mutation or phosphorylation that leads to its aggregation, affecting the formation and coupling of lysosomal vesicles without producing lysosomal autophagy, and leading to its deposition in dopaminergic neurons, the formation of LBs and apoptosis [49].

The results presented here show that, among the proteins deregulated in the tear fluid of PD patients, both iPD patients and carriers of the E46K-SNCA mutation, there are two proteins related to lysosomal autophagy: CATD and ASAH1. CATD is an important lysosomal aspartyl protease, and, interestingly, CATD deficiency and its enzymatic inactivation in humans result in an early onset of progressive and eventually fatal neurodegeneration, which is classified as one of several neuronal ceroid lipofuscinosis (NCL) syndromes [50,51]. In vitro, CATD produces partial proteolysis of recombinant α-syn [52], and the ability of CATD to regulate α-syn has been evaluated in wild-type and mutant dopaminergic cells in culture. Moreover, the brains of several CATD-deficient mammals with NCL were studied to assess endogenous α-syn processing, and, consequently, it was concluded that the enzymatic activity of CATD plays an important role in the metabolism of α-syn.

The ASAH1 protein is a lysosomal enzyme that converts lysosomal ceramide to sphingosine. Inhibition of ASAH1 increases ceramide levels, also reducing the amount of oxidized α-syn and that of the ubiquitinylated protein in dopaminergic neurons derived from PD patients. A decrease in the levels of ceramide due to enhanced ASAH1 activity may contribute to the accumulation of intracellular α-syn, since it alters lysosomal autophagy, perhaps inhibiting the release of α-syn to the extracellular space.

DYHC1 and LMNA are two other relevant proteins deregulated in the tears of patients with PD. DYHC1 is a protein involved in the movement of organelles within the cell and retrograde transport in axons. Among other neurodegenerative diseases, it is implicated in AD and PD. The alterations to DYHC1 cause poor transport to its normal synaptic location and poor protein clearance, such as that of the lysosomal protease cathepsin D [53]. This proposed sequence of events would generate an autocatalytic spiral in which the processes that lead to the inhibition of axonal transport and the production of phosphorylated α-syn become mutually stimulating, providing a rational explanation for the early synaptic loss in the striatum of PD patients. Finally, the overexpression of the LMNA protein in the PD tear has been implicated in demyelination, yet this has not been related to PD to date [54]. However, this is a route that deserves to be considered in future studies.

This study has some limitations, such as the low number of patients included, so it should be considered as a pilot study, and this should be taken into account when interpreting the results. In addition, it would be necessary to validate the candidate proteins, not only in a larger cohort, but also to check if they appear in other body fluids (i.e., blood or cerebrospinal fluid). It is also important to keep in mind that the possible mechanisms proposed in this article are only hypotheses that would need to be verified in further studies. In addition, it should be noted that, since the database searches took into account canonical protein sequences, no information on proteoforms was obtained. Although this could represent a limitation in the specificity of the results, we believe that the information obtained at the canonical sequence level is still of great value and provides valuable insights into the molecular characteristics of the disease. However, the possibility that certain proteoform(s) may be responsible for the annotated results should be kept in mind when interpreting the data.

## 5. Conclusions

In this pilot study, a proteomic analysis revealed that certain proteins were upregulated in the tears of PD patients, mainly those involved in lysosomal function. The importance of this study in identifying proteins in the tear involved in neurodegeneration should be highlighted and their relationship to PD patients with an aggressive disease phenotype. Since this was a pilot study, only a limited number of patients were studied. A limitation of the study is the difference in age between the control group and the PD group. Analysis with tear samples in healthy subjects without ocular pathologies is complicated since, at a certain age, it is difficult to find volunteers without alterations in the tear film. Future trials with larger patient cohorts will allow us to identify specific biomarkers for PD that would ideally help to predict the onset of this disease. Nevertheless, we provide evidence here that the proteome of individual patients can be analyzed by using only a limited amount of tears. With good training, the best way to extract tears for proteomic studies is using a glass capillary, thereby avoiding the cell contamination that may occur when Schirmer’s strips are used. In future studies, we hope to validate the results presented here and identify endophenotypes of PD through the tear proteomic profile, which could serve to achieve the early diagnosis of PD.

## Figures and Tables

**Figure 1 proteomes-10-00004-f001:**
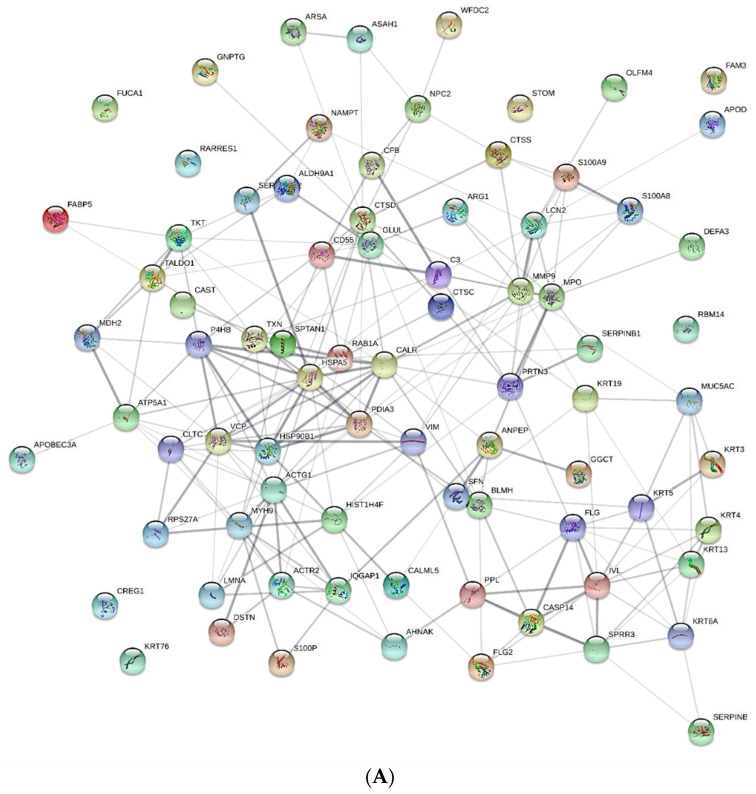
A STRING PPI network analysis showing the direct interactions of the biomarkers identified. The associations between the proteins shown are intended to be specific and significant, such that the proteins that interact participate in a shared activity. Splicing isoforms or post-translational modifications collapse, such that each node represents all proteins produced by a single protein-coding gene locus. (**A**) Protein–protein interaction (PPI) network of the proteins upregulated in PD patients relative to the CTs. (**B**) PPI network of the proteins downregulated in the tear fluid of PD patients.

**Figure 2 proteomes-10-00004-f002:**
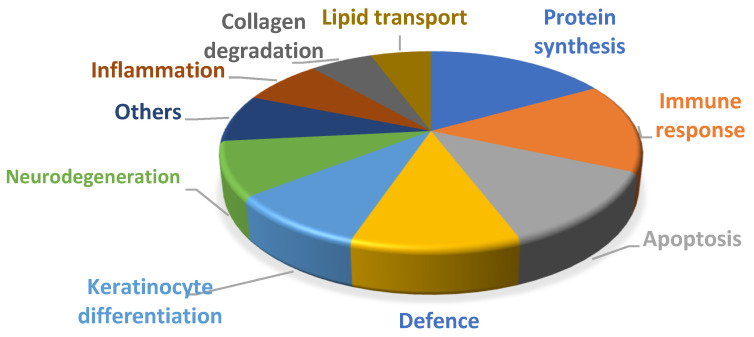
Relative biological functions of the deregulated proteins identified in PD tears expressed as a %.

**Figure 3 proteomes-10-00004-f003:**
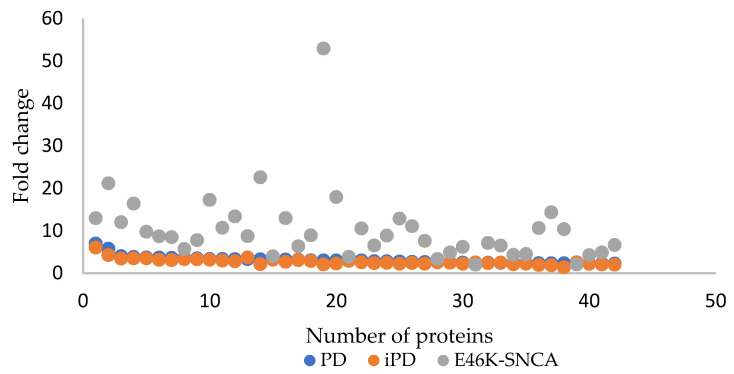
Representation of the fold change in expression of the significant proteins when the patients included in the study were considered as iPD patients (orange), E46K-SCNA carriers (gray) or all PD patients together (blue). The same deregulated proteins appear in all three groups, but the changes in expression were stronger in the group of E46K-SCNA carriers.

**Table 1 proteomes-10-00004-t001:** Demographics, PD characteristics and eye conditions.

Variables, Units	iPD	E46K-SNCA	Control
*n*	24	3	27
Age, years	60.4 (9.21)	45 (14.73)	49.69 (12.17)
Gender (F/M) (%)	38/62	0/100	54/46
Disease duration, years	9.36 (6.66)	3.33 (3.51)	N/A
Hoehn and Yahr score	2 (0.82)	2 (1)	N/A
UPDRS score (%)	32.39 (9.03)	16.56 (11.12)	N/A
Part I UPDRS score	17.19 (12.6)	6.25 (0.0)	N/A
Part II UPDRS score	27.52 (10.07)	11.54 (8.38)	N/A
Part III UPDRS score	45.13 (15.09)	25.00 (16.7)	N/A
Part IV UPDRS score	16.30 (9.46)	10.14 (9.05)	N/A
Blepharitis (%)	40.74	0	0

The results are displayed as the means (standard deviation), except for the sex and Hoehn and Yahr scores that are shown as the percentage and median: UPDRS, Unified Parkinson’s Disease Rating Scale; *n*, number of subjects; F, female; iPD, idiopathic Parkinson’s disease; M, male.

**Table 2 proteomes-10-00004-t002:** Deregulated proteins comparing iPD with blepharitis vs. iPD without blepharitis.

Entry Name	Protein Name	*p*-Value	Fold
OLFM4	Olfactomedin-4	7.08 × 10^−5^	4.25
SPB3	Serpin B3	5.29 × 10^−4^	3.99
SPRR3	Small proline-rich protein 3	1.77 × 10^−2^	3.31
MMP9	Matrix metalloproteinase-9	7.33 × 10^−3^	3.08
CASPE	Caspase-14	4.88 × 10^−2^	2.67
PRTN3	Myeloblastin	1.76 × 10^−2^	2.63
GGCT	Gamma-glutamylcyclotransferase	2.56 × 10^−2^	2.40
CATD	Cathepsin D	1.24 × 10^−3^	2.31
CALL5	Calmodulin-like protein 5	3.48 × 10^−2^	1.91
TIG1	Retinoic acid receptor responder protein 1	3.87 × 10^−2^	1.90
MDHM	Malate dehydrogenase mitochondrial	8.59 × 10^−3^	1.66
NGAL	Neutrophil gelatinase-associated lipocalin	3.51 × 10^−2^	1.54
NQO1	NAD(P)H dehydrogenase (quinone) 1	1.92 × 10^−2^	0.49
PLST	Plastin-3	1.15 × 10^−2^	0.40
MYH14	Myosin-14	4.80 × 10^−2^	0.37
AMPL	Cytosol aminopeptidase	3.78 × 10^−2^	0.36
AK1A1	Alcohol dehydrogenase (NADP(+))	3.53 × 10^−2^	0.35
ADH1G	Alcohol dehydrogenase 1C	2.64 × 10^−3^	0.35
PRDX5	Peroxiredoxin-5 mitochondrial	4.71 × 10^−2^	0.35
RINI	Ribonuclease inhibitor	2.21 × 10^−2^	0.34
VATA	V-type proton ATPase catalytic subunit A	1.49 × 10^−4^	0.25

Note: *p* < 0.05 statistical significances; fold change > 1.5 for upregulated proteins; fold change < 0.5 for downregulated proteins.

**Table 3 proteomes-10-00004-t003:** Deregulated tear proteins related to neurodegenerative functions.

Entry Name	Protein Name	Peptides	Unique Peptides	*p*-Value	Fold Change	AUC %
LMNA	Prelamin-A/C	25	24	4.00 × 10^−2^	2.25	67.1
CATD	Cathepsin D	19	19	1.48 × 10^−3^	1.82	75.4
ASAH1	Acid ceramidase	6	6	2.98 × 10^−3^	1.80	72.3
TERA	Transitional endoplasmic reticulum ATPase	18	18	1.99 × 10^−2^	1.60	65
DYHC1	Cytoplasmic dynein 1 heavy chain 1	9	9	7.50 × 10^−3^	1.32	70.2
TPP1	Tripeptidyl-peptidase 1	11	11	4.13 × 10^−2^	0.64	69

AUC = area under the ROC curve.

**Table 4 proteomes-10-00004-t004:** Univariate logistic regression models.

Unique Variable Used in Model	Estimate	*p*-Value
Sex	1.163	0.072
Age	0.088	0.00 *
LMNA	0.351	0.055
CATD	1.426	0.010 *
ASAH1	0.990	0.020 *
TERA	0.801	0.040 *
DYHC1	1.197	0.085
TPP1	−0.527	0.147
Time with disease	9.483	0.995
UPDRS score	488.104	0.995
Part I UPDRS score	311.136	0.994
Part II UPDRS score	964.522	0.995
Part III UPDRS score	313.201	0.995
Part IV UPDRS score	398.234	0.995
HY Score	20.013	0.995

Note: * *p* < 0.05 statistical significance. Part UPDRS, the Unified Parkinson’s disease rating scale’s domains.

**Table 5 proteomes-10-00004-t005:** Multivariate logistic regression model.

Multivariate Model Variables	Estimate	*p*-Value
(Intercept)	−50.576	0.053
Age	0.166	0.040 *
LMNA	0.495	0.181
CATD	1.670	0.040 *
ASAH1	0.141	0.912
TERA	0.470	0.655
DYHC1	0.798	0.602

Note: * *p* < 0.05 statistical significances.

## Data Availability

The proteomic data obtained are available at ProteomeXchange with identifier 10.6019/PXD028811.

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
