# Peer review of "Potential Tear Biomarkers for the Diagnosis of Parkinson’s Disease—A Pilot Study"

_proteomes, 2022, doi:10.3390/proteomes10010004_

Round 1
Reviewer 1 Report
An interesting way of a non invasive collection of tear samples for proteomic studies that could be relevant for not only PD but neurodegenerative diseases (NDD) in general.
Because the collection method is at the center of the presentation, it has to be noted that having glass tubes so close to the eye would imply special training, potential conflicts and very specific disclaimers. It occurred to this reviewer that some conflicts could be resolved by adding at the top a 0.5 cm piece of thin wall silicone tubing of the same diameter as the glass capillary tube and protruding not more than 2 mm; an alternative that could fit most users of different trainings. This is a suggestion and has no bearing on the judgment of presentation.
Materials and Methods would benefit with better clarifications of 2.2 and 2.3; there are different topics, ophthalmologic and PD scores, that not everybody would be familiarized; specially to understand the results presented in table 1.
The quality of the figures, specially figures 1 and 2, is unacceptable; impossible to see even magnifying at 250% in a big screen. Definition has to be definitely improved.
As a note in figure 1 the position of ApoE (strongest link to AD) merits at least a few words.
Because of the emphasis biomarkers linked to the immune system and inflammation; this reviewer thinks the data on iPD should be broken into subjects positive and negative for blepharitis (a definite sign of inflammation and potential autoimmune involvement)
Also there should be some word about the proteins that are involved in the maintenance or modification of the ECM (extra cellular matrix).
It is true that some of the upregulated proteins would have an effect in the accumulation of α-syn; however, there are possible links to the involvement (i.e., cathepsin D and others) in the conformational change that would render pathologic the oligomers and precipitates of α-syn. May be a few words about conformation in iPD would be necessary.
Otherwise, it is an excellent beginning for a most needed development in the quest for early biomarkers in NDD, without the need of collecting biological fluids using invasive techniques.
Author Response
R1: An interesting way of a noninvasive collection of tear samples for proteomic studies that could be relevant for not only PD but neurodegenerative diseases (NDD) in general.
Because the collection method is at the center of the presentation, it has to be noted that having glass tubes so close to the eye would imply special training, potential conflicts and very specific disclaimers. It occurred to this reviewer that some conflicts could be resolved by adding at the top a 0.5 cm piece of thin wall silicone tubing of the same diameter as the glass capillary tube and protruding not more than 2 mm; an alternative that could fit most users of different pieces of training. This is a suggestion and has no bearing on the judgment of presentation.
A: It's a great idea. Thank you for the suggestion, we are going to test this tear collection method in future studies starting with the validation of the present work.
R1: Materials and Methods would benefit from better clarifications of 2.2 and 2.3; there are different topics, ophthalmologic and PD scores, that not everybody would be familiarized with; especially to understand the results presented in table 1.
A: The tests carried out have been explained in more detail in the text.
R1: The quality of the figures, especially figures 1 and 2, is unacceptable; impossible to see even magnifying at 250% on a big screen. The definition has to be definitely improved.
A: Following the suggestion that you have proposed, new figures have been generated with higher resolution and with a more understandable structure according to what we want to show.
R1: As a note in figure 1 the position of ApoE (strongest link to AD) merits at least a few words.
A: Thanks for the suggestion. A section about APO family proteins has been included in the Discussion (page 12).
R1: Because of the emphasis on biomarkers linked to the immune system and inflammation; this reviewer thinks the data on iPD should be broken into subjects positive and negative for blepharitis (a definite sign of inflammation and potential autoimmune involvement)
A: The analyses have been carried out separating all the groups. It has been compared by differentiating the type of parkinsonism, idiopathic, or presenting the E46K-SNCA mutation (Figure 2) and within patients with iPD (patients with the mutation did not present blepharitis), iPD positive blepharitis, and negative blepharitis (Table 2).
R1: Also there should be some word about the proteins that are involved in the maintenance or modification of the ECM (extracellular matrix).
A: ECM and MMPs involvement are commented on in the discussion (page 11)
R1: It is true that some of the upregulated proteins would have an effect in the accumulation of α-syn; however, there are possible links to the involvement (i.e., cathepsin D and others) in the conformational change that would render pathologic the oligomers and precipitates of α-syn. Maybe a few words about conformation in iPD would be necessary.
A: This has been included in the discussion section (page 12)
R1: Otherwise, it is an excellent beginning for a most needed development in the quest for early biomarkers in NDD, without the need of collecting biological fluids using invasive techniques.
A: Thank you very much.

Reviewer 2 Report
The study of Acera et al is focused on important problem of finding Parkinson's disease biomarkers. PD diagnostics in early stage patients is problematic and what is even more important reliable methods of establishing PD diagnosis in presymptomatic patients are absent. Importantly, in patients with present motor symptoms significant portion of nigrostriatal dopamine neurons is lost and therefore potential neuroprotective treatments fail in clinical trials. Therefore, the research focused on finding biomarkers for early PD diagnosis has high value and such data are very much in demand in clinics.
The study of Acera et al although interesting has some limitations, 2 of which are the most important:
- The iPD patients selected for the study have a disease duration of more than 9 years in average, which corresponds to moderate rather than to early stage PD. The authors also had a small group of PD patients with seemingly genetic early onset PD, but from the data presentation it is unclear if the data collected from these patients were analyzed separated or were pooled with the data of iPD patients. Moreover, patients with genetic PD is rather a minor group and it is unclear if such data can be extrapolated to the majority of patients who actually have iPD. Therefore, conclusions concerning the options for potential early PD diagnostics seem to be unjustified. I suggest the authors to reconsider the concept of the article and switch the focus from early diagnostics to PD biomarkers which is much more in line with their data. Also it will be interesting to see the data for genetic PD patients analyzed separately and compared to the data for iPD.
- The groups seem to be unbalanced: the age of patients in the control group is almost 11 years lower than the age of patients in iPD group. Also a number of PD patients had blepharitis. Since many of the proteins identified by the authors as dysregulated are related to cancer (age-related disease), cell damage and inflammation, these factors should be considered and their (age and blepharitis) contribution should be demonstrated and discussed.
The data presentation is not adequate:
Table 1 – please check the formatting of the text below the table. It is unclear from the first glance, that it belongs to the Table 1 legend
Figure 1, Figure 2A – labels in the figures are too small and unreadable
Supplemental information: it is unclear what do you present in this file, please, clarify the column labels.
I also have in mind a few minor issues to be addressed by authors: 1. Patients exclusion and inclusion criteria should be described more clearly. 2. In some places stylistic language changes are needed. 3. Some typos are spotted.
Examples (the list does not contain all issues) Line 139. Typo: 400 nL/min. the column > 400 nL/min.
Line 144. Please, define FDR if not defined earlier
Line 150-154. Please, rephrase, difficult to understand
Author Response
R2: The study of Acera et al is focused on important problem of finding Parkinson's disease biomarkers. PD diagnostics in early stage patients is problematic and what is even more important reliable methods of establishing PD diagnosis in presymptomatic patients are absent. Importantly, in patients with present motor symptoms significant portion of nigrostriatal dopamine neurons is lost and therefore potential neuroprotective treatments fail in clinical trials. Therefore, the research focused on finding biomarkers for early PD diagnosis has high value and such data are very much in demand in clinics.
The study of Acera et al although interesting has some limitations, 2 of which are the most important:
- The iPD patients selected for the study have a disease duration of more than 9 years on average, which corresponds to moderate rather than early-stage PD. The authors also had a small group of PD patients with seemingly genetic early-onset PD, but from the data presented it is unclear if the data collected from these patients were analyzed separated or were pooled with the data of iPD patients. Moreover, patients with genetic PD are rather a minor group and it is unclear if such data can be extrapolated to the majority of patients who actually have iPD. Therefore, conclusions concerning the options for potential early PD diagnostics seem to be unjustified. I suggest the authors to reconsider the concept of the article and switch the focus from early diagnostics to PD biomarkers which is much more in line with their data. Also, it will be interesting to see the data for genetic PD patients analyzed separately and compared to the data for iPD.
A: Now, the analyses have been carried out separating all the groups. It has been compared by differentiating the type of parkinsonism, idiopathic, or presenting the E46K-SNCA mutation (Figure 2) and within patients with iPD (patients with the mutation did not present blepharitis), iPD positive blepharitis, and negative blepharitis (Table 2).
We remove the “early” term in the main part of the manuscript and in the title.
It is true that in this study most of the patients have a mean duration of the disease of 9 years and patients with an early diagnosis have not been chosen. We will have to test the candidates for biomarkers that will be validated and conclusive in the tears of patients with early stages of the disease and their relatives, even if they do not develop the disease, it can be seen if there is a tendency or not to develop it. The family with the E46K-SNCA mutation at younger ages develops the disease. When separating the samples from these patients in the analyses, it is observed that the overexpression of the proteins is greater in the case of patients with the mutation correlated with the severity of the pathology. Currently, there are only 8 family members with PD. The selected and validated candidates will be analyzed by all family members. It is true that at this time we cannot say that biomarker candidates are diagnosed early.
- The groups seem to be unbalanced: the age of patients in the control group is almost 11 years lower than the age of patients in iPD group. Also a number of PD patients had blepharitis. Since many of the proteins identified by the authors as dysregulated are related to cancer (age-related disease), cell damage, and inflammation, these factors should be considered and their (age and blepharitis) contribution should be demonstrated and discussed.
A: We have indicated at the end of the manuscript that a limitation of the study is the difference in age between the control group and the PD group. Analysis with tear samples in healthy subjects without ocular pathologies is complicated since at a certain age it is difficult to find volunteers without alterations in the tear film. (Page 13)
The analyses have been carried out separating the patients with and without blepharitis as requested (Table 2). The proteins that have been proposed as possible candidate markers are proteins related to neurodegenerative diseases and not related to ocular surface pathologies, precisely because of the existence of blepharitis in Parkinson's patients.
R2: The data presentation is not adequate:
-Table 1 – please check the formatting of the text below the table. It is unclear from the first glance, that it belongs to the Table 1 legend
A: We checked the formatting of the text
-Figure 1, Figure 2A – labels in the figures are too small and unreadable
A: Following the suggestion that you have proposed, new figures have been generated with higher resolution and with a more understandable structure according to what we want to show.
-Supplemental information: it is unclear what do you present in this file, please, clarify the column labels.
A: We included in legends the column information:
Groups: Sample grouping based on different categories. Group 3 was used for the Ctrl vs PD main comparison.
Protein identification FDR1%: Proteins identified with a peptide-level FDR1%. Accession and Description provide protein information as contained in the Uniprot database. -10logP accounts for the identification score. Sequence coverage % represents the % of the total protein sequence that gets covered with the peptides identified in our experiment. The area is the extracted-ion chromatogram (XIC) are calculated by the software.
#Peptides is the total number of peptides identified for that protein, whereas #Unique is the number of peptides that match exclusively to that protein (not shared with any other protein identified in the dataset). Spectral counts are the total number of spectra identified for a protein. PTM describes the post-translational modifications identified for a protein and Avg. Mass corresponds to the protein mass in Da.
Protein Quantification: This sheet contains protein quantification data (Area) obtained from the PEAKS Q module. This information was used for the differential analysis.
Differential analysis results: data obtained after quantitative data processing and statistical analysis.
Columns I and J contain the Student’s t-test p-value and the ratio, whereas L-BM contains protein area after data processing in Perseus (described in Materials and Methods).
I also have in mind a few minor issues to be addressed by authors:
- Patients' exclusion and inclusion criteria should be described more clearly.
A: More information has been included in these sections
- In some places, stylistic language changes are needed.
A: English has been revised by a native scientific translator
- Some typos are spotted.
Examples (the list does not contain all issues) Line 139. Typo: 400 nL/min. the column > 400 nL/min.
Line 144. Please, define FDR if not defined earlier
A: We have included in the text: False Discovery Rate (FDR)
Line 150-154. Please, rephrase, difficult to understand
A: It is the way in which the proteomes journal tells us that we should indicate the supplementary information. It is a database where the journal has asked us to upload all the raw data from the proteomics analyzes so that there is free access to the information. We have to indicate that reference like this in the methods section.

Reviewer 3 Report
The study of Acera et al. analyses tear proteome as a possible source of Parkinson´s disease biomarkers. Tears have been previously subject of studies in other neurodegenerative diseases, mainly due to their non-invasive collection. The presented study is novel as it analyses individual patients´ samples. One previous study analyzed tear proteome in pooled tear samples of PD patients (Boerger et al. 2019) and additional one study quantified a-synuclein levels in tears (Maass 2020).
Despite the results are interesting for publication, there are several issues that should be covered.
1) It is not clear, how was the quantification done in the proteomic analysis. Add the description.
2) The description of results of the protein-protein interaction study by STRING provides only very limited information. Do the proteins, for example, cluster, based on their interactions, into any functional groups/pathways? In the Fig. 1A, there seem to be some functional groups (e.g. structural proteins – actin isoforms, vimentin, keratins; immune response regulation – S100A9, CD55; metabolic enzymes… etc.). In the Fig 1B, it seems that the lipid transport/metabolism proteins (e.g. several lipoproteins and their receptors) are downregulated in the patients´ tears. In contrast, the proteomic analysis of tears in PD patients done by Boerger et al. 2019, showed increased ApoD in PD patients´ tears. It should be discussed if such lipid changes have been previously found (e.g. in blood plasma) of PD patients. Such proteins may also represent leakage from blood plasma into the tears and might be detected as significant just by a chance due to lower number of analyzed individuals.
3) Describe more clearly what the logistic regression analysis shows, i.e. which event of interest was modeled. Is it the chance of having PD?
4) There are many typing errors and unclear information in the text that need to be corrected:
Line 24 and Line 82: term “candidate biomarker” should be used instead of “biomarker” as only limited number of samples was analyzed and the results were not verified in larger cohort.
Line 87: State more clearly, that 54 subjects were the total number, from which 24 were patients with iPD, 3 carriers of the E46K-SNCA mutation and 27 CTs.
Line 139: (T in capitals) The column was heated
Lines 150-151: The text “The PRIDE database and related tools and resources in 2019: improv-150 ing support for quantification data. Nucleic Acids Res 47(D1):D442-D450 (PubMed ID: 151 30395289).)” was inserted probably by a mistake (it is a reference).
Line 159: Correct the text p-value<05. Should be p-value<0.05.
Lines 161-163: the authors describe dementia, however, the dementia was not subject of the presented study. Correct it.
Line 172 – Table 1: It is not clear what means the Dimension I, II, III, IV.
Line 176: remove “SD: Standard“
Fig 1A: Provide the figure with higher resolution.
Fig 2: provide tee figure with higher resolution and increase font size. In the Fig 2 legend, remove the word “markers“, and use just “protein“, as the markers need deeper validation and verification. The label “FOLD (PD vs CT)“ belongs to the Fig 2A, not to the Fig 2B.
Line 226 and Table 3: There is discrepancy in the statistical significance of the proteins TERA and DYHC1 between Line 226 and table 3. The abbreviation HY in Table 3 (last line) should be explained.
Author Response
R3: The study of Acera et al. analyses tear proteome as a possible source of Parkinson´s disease biomarkers. Tears have been previously subject of studies in other neurodegenerative diseases, mainly due to their non-invasive collection. The presented study is novel as it analyses individual patients´ samples. One previous study analyzed tear proteome in pooled tear samples of PD patients (Boerger et al. 2019) and additional one study quantified a-synuclein levels in tears (Maass 2020).
Despite the results are interesting for publication, there are several issues that should be covered.
- It is not clear, how was the quantification done in the proteomic analysis. Add the description.
A: It has been added in the methods section: Protein identification and quantification was carried out using the PEAKS software (Bioinformatics solutions). Searches were carried out against a database of canonical human Uniprot/Swissprot entries, with precursor and fragment tolerances of 20 ppm and 0,05 Da. An area-based label-free protein quantification was performed using the PEAKS Q module available in PEAKS software.
- The description of the results of the protein-protein interaction study by STRING provides only very limited information. Do the proteins, for example, cluster, based on their interactions, into any functional groups/pathways? In Fig. 1A, there seem to be some functional groups (e.g. structural proteins – actin isoforms, vimentin, keratins; immune response regulation – S100A9, CD55; metabolic enzymes… etc.). In the Fig 1B, it seems that the lipid transport/metabolism proteins (e.g. several lipoproteins and their receptors) are downregulated in the patients´ tears. In contrast, the proteomic analysis of tears in PD patients done by Boerger et al. 2019, showed increased ApoD in PD patients´ tears. It should be discussed if such lipid changes have been previously found (e.g. in blood plasma) of PD patients. Such proteins may also represent leakage from blood plasma into the tears and might be detected as significant just by a chance due to lower number of analyzed individuals.
A: It has been explained in the discussion section more about different families of proteins that are deregulated in the present study. Thus, MMPs and APO proteins have been included in the discussion section. The explanation of the interaction between proteins provided by STRING has been explained at the bottom of the figure. More restrictive analyzes have been performed in which the interactions between the study proteins appear more clearly.
3) Describe more clearly what the logistic regression analysis shows, i.e. which event of interest was modeled. Is it the chance of having PD?
A: Indeed, the logistic models establish the effect that the different covariates have on the possibility that the patient suffers PD. In methods, the description of the logistic models is expanded to indicate the desired result variable.
4) There are many typing errors and unclear information in the text that need to be corrected:
-Line 24 and Line 82: the term “candidate biomarker” should be used instead of “biomarker” as only a limited number of samples was analyzed and the results were not verified in a larger cohort
A: They have been changed in the manuscript. Thank you for the comment.
-Line 87: State more clearly, that 54 subjects were the total number, from which 24 were patients with iPD, 3 carriers of the E46K-SNCA mutation, and 27 CTs.
A: We removed 54 subjects (total subjects included in the study)
-Line 139: (T in capitals) The column was heated
A: Thank you. It has been changed
-Lines 150-151: The text “The PRIDE database and related tools and resources in 2019: improv-150 ing support for quantification data. Nucleic Acids Res 47(D1):D442-D450 (PubMed ID: 151 30395289).)” was inserted probably by a mistake (it is a reference).
A: It is the way in which the proteomes journal tells us that we should indicate the supplementary information. It is a database where the journal has asked us to upload all the raw data from the proteomics analyzes so that there is free access to the information. We have to indicate that reference like this in the methods section.
-Line 159: Correct the text p-value<05. Should be p-value<0.05.
A: It has been corrected in the text
-Lines 161-163: the authors describe dementia, however, dementia was not the subject of the presented study. Correct it.
A: The term dementia refers to Parkinson's disease. The term was modified to avoid errors.
-Line 172 – Table 1: It is not clear what means the Dimension I, II, III, IV.
A: Dimensions I to IV refer to the different parts of the UPDRS test (Unified Parkinson's disease rating scale). A table footer has been included to avoid confusion. The UPDRS scale is a classification system designed for longitudinal monitoring of the course of PD. The scale was developed by a committee directed by Professor Stanley Fahn in 1984. It is an instrument composed of the following domains: part I: mental, behavioral, and mood; part II: activities of daily living; part III: motor evaluation; and part IV: motor complications.
Fahn S, Elton R, Members of the UPDRS Development Committee. In: Fahn S, Marsden CD, Calne DB, Goldstein M. Eds. Recent developments in Parkinson’s disease, Florham Park, NJ. Macmillan Health Care Information 1987;(2)153-163:293-304.
-Line 176: remove “SD: Standard“
A: It has been removed
-Fig 1A: Provide the figure with higher resolution.
A: Following the suggestion that you have proposed, new figures have been generated with higher resolution and with a more understandable structure according to what we want to show.
-Fig 2: provide tee figure with higher resolution and increase font size. In the Fig 2 legend, remove the word “markers“, and use just “protein“, as the markers need deeper validation and verification. The label “FOLD (PD vs CT)“ belongs to Fig 2A, not to Fig 2B.
A: Following the suggestion that you have proposed, new figures have been generated with higher resolution and with a more understandable structure according to what we want to show.
-Line 226 and Table 3: There is a discrepancy in the statistical significance of the proteins TERA and DYHC1 between Line 226 and table 3. The abbreviation HY in Table 3 (last line) should be explained.
A: The difference between the p-values corresponds to the difference in the statistical techniques used: The estimation of the discrimination capacity through the direct use of a continuous variable (ROC curve) and the use of logistic regression models. The ROC curve describes the quality of the binary classification ability of a variable. The measure is based on factoring in the base likelihood ratio according to Murphy and Winkler's (1987) formalism. The procedure establishes the correspondence between the prognoses (sick or healthy patient) and the real observations. This correspondence is described by a two-component vector defined by the 'hit rate' (the proportion of events for which PD diagnosis was correctly issued) and the 'miss rate' (the proportion of non-events for PD diagnosis). The hit rate and the miss rate establish a (two-dimensional) coordinate in the ROC space and collectively define the ROC curve.
The Significance level or P-value is the probability that the observed sample Area under the ROC curve is found when in fact, the true (population) Area under the ROC curve is 0.5 (null hypothesis: Area = 0.5). Thus, if the p-value is small (P <0.05), then it can be concluded that the Area under the ROC curve is significantly different from 0.5 and that is evidence that the variable considered has the ability to distinguish between the two groups (Hanley & McNeil, 1982; Zweig & Campbell, 1993).
Logistic models allow estimating the relationship between a discrete dependent variable (patients with or without PD) and a set of continuous or discrete independent variables. Thus, logistic regression is a type of regression analysis used to predict the outcome of a categorical variable based on the independent or predictor variables. It is useful for modeling the probability of an event occurring as a function of other factors. The logistic regression analysis is part of the set of Generalized Linear Models (GLM) that uses the logit function as the link function. In the models, the p-value associated with each variable tests the null hypothesis that the coefficient of the regression model is equal to zero. That is, the variable has no effect on the binary result considered. Thus, a low p-value (<0.05) indicates that the null hypothesis can be rejected, there is a high probability that the changes in the predictor value are related to changes in the response variable. However, the p-values do not indicate the classification quality of the model in the sample considered).
Thus, the difference between the p-values of the AUC of the ROC curves and the p-values of the logistic models analyze different measures of classification capability. Despite the fact that the two measures analyze, from different points of view, the classification capacity of the predictor variables.
Hanley, J. A. and McNeil, B. J. 1982 The meaning and use of the area under the receiver operating characteristic (ROC) curve. Radiology, 143, 29–36
Campbell, G. 1994 Advances in statistical methodology for the evaluation of diagnostic and laboratory tests. Stat. Med., 13, 499–508

Round 2
Reviewer 1 Report
The authors have completed a thorough examination and changes in the manuscript, addressing all the reviewers concerns. Figures are really improved, although some of the graphs are not at an optimal definition yet. Nevertheless, blowing in the computer one can read and differentiate all the proteins analyzed.
Now the manuscript and the discussion is more interesting and would benefit the audience and the author's value in their research and findings.
One more suggestion. In the future try to explore why if the T-cell and inflammation involvement is modified in PD; the IgA2 (no disulfide bridge between H and L and a specific conformation to survive in secretions like tears) is not modified at least in concentration (supplementary info). Mucosal and epithelial immunoglobulins are extremely sensitive to changes in their domains, more when they involve a self conformational modified protein.
Author Response
The authors have made a thorough review and changes to the manuscript, addressing all the reviewers' concerns. The figures are much improved, although some of the graphs are still not optimally defined. However, by blowing into the computer all the proteins analyzed can be read and differentiated.
We very much appreciate the comment and hope that in the publication the figure size can be maximized without losing resolution.
Now the manuscript and discussion is more interesting and would benefit the audience and the author's value in his research and findings.
Yes, we are very grateful for the recommendations made by the reviewers.
One more suggestion. In the future try to explore why if T cell involvement and inflammation is modified in PD; IgA2 (no disulfide bridge between H and L and a specific conformation to survive in secretions such as tears) is not modified at least in concentration (supplementary information). Mucosal and epithelial immunoglobulins are extremely sensitive to changes in their domains, more so when it is a self-conforming protein.
Thank you very much for the recommendation. In the validation study, we will specifically analyze the concentration of IgA2 in tears to evaluate any change in concentration. In our study, no statistically significant changes are observed, it may be that the volume of tear analyzed does not allow us to see these changes if the concentration is very low. We will also analyze this protein by quantitative techniques such as ELISA, to see if analyzing the protein in isolation and in a larger volume of tear, we see any significant change.

This manuscript is a resubmission of an earlier submission. The following is a list of the peer review reports and author responses from that submission.